# Predicting Wins, Losses and Attributes’ Sensitivities in the Soccer World Cup 2018 Using Neural Network Analysis

**DOI:** 10.3390/s20113213

**Published:** 2020-06-05

**Authors:** Amr Hassan, Abdel-Rahman Akl, Ibrahim Hassan, Caroline Sunderland

**Affiliations:** 1Department of Sports Training, Faculty of Sports Education, Mansoura University, Mansoura 35516, Egypt; 2Faculty of Physical Education-Abo Qir, Alexandria University, Alexandria 21913, Egypt; abdelrahman.akl@alexu.edu.eg; 3Faculty of Physical Education, Zagazig University, Zagazig 44519, Egypt; ibrahimhassan@fopem.zu.edu.eg; 4Department of Sport Science, Sport, Health and Performance Enhancement Research Centre, School of Science and Technology, Nottingham Trent University, Nottingham NG11 8NS, UK; caroline.sunderland@ntu.ac.uk

**Keywords:** football, match analysis, neural network, sport analytics

## Abstract

Predicting the results of soccer competitions and the contributions of match attributes, in particular, has gained popularity in recent years. Big data processing obtained from different sensors, cameras and analysis systems needs modern tools that can provide a deep understanding of the relationship between this huge amount of data produced by sensors and cameras, both linear and non-linear data. Using data mining tools does not appear sufficient to provide a deep understanding of the relationship between the match attributes and results and how to predict or optimize the results based upon performance variables. This study aimed to suggest a different approach to predict wins, losses and attributes’ sensitivities which enables the prediction of match results based on the most sensitive attributes that affect it as a second step. A radial basis function neural network model has successfully weighted the effectiveness of all match attributes and classified the team results into the target groups as a win or loss. The neural network model’s output demonstrated a correct percentage of win and loss of 83.3% and 72.7% respectively, with a low Root Mean Square training error of 2.9% and testing error of 0.37%. Out of 75 match attributes, 19 were identified as powerful predictors of success. The most powerful respectively were: the Total Team Medium Pass Attempted (MBA) 100%; the Distance Covered Team Average in zone 3 (15–20 km/h; Zone3_TA) 99%; the Team Average ball delivery into the attacking third of the field (TA_DAT) 80.9%; the Total Team Covered Distance without Ball Possession (Not in_Poss_TT) 76.8%; and the Average Distance Covered by Team (Game TA) 75.1%. Therefore, the novel radial based function neural network model can be employed by sports scientists to adapt training, tactics and opposition analysis to improve performance.

## 1. Introduction

The continual innovation, development and use of technology within sport has resulted in large data sets from soccer. World Cup matches are available to analyze and explain player’s and team’s performance. The increased accessibility of detailed match data is largely due to the progression made in semi- and fully-automated tracking [1,2,3,4]. Noticeably, the technical and tactical match performance of professional soccer players, has been shown to be affected by several different contextual, situational and positional attributes [5,6,7,8,9,10]. Thus, the continuous movement dynamic and the corresponding movement within the context of the match produces thousands of different values that reflect in total what the player, players, team or teams do together as a whole during the game. 

In the last decade, there has been a plethora of research describing and analyzing the match performance data from soccer World Cups using several approaches such as multivariate analyses and machine learning [1,2,3,4,11,12,13,14,15], passing networks based on space, time and the multilayer nature of the game [16] or based on spatial and temporal entropy related to football teams and their players by means of a pass-based interaction [17] and social network analyzes to study the interaction between a player and their teammates (for example a ball passing network) through graph theory to assess the structural and topographical characteristics of personal interactions between team members [18]. This type of descriptive research provides important information that can be used to improve training and adapt tactics, however analyses, such as machine learning, can identify performance indicators, whether physical or technical that may predict what will occur during the match [19,20,21,22]. Furthermore, for performance attributes and events to be meaningful, they need to be relevant to the team’s success such as the winning of matches [23]. Such analysis needs an amount of data which is an outcome from different sensors, cameras and analysis systems and needs an intelligent solution to invest in these big data which are usually linear and nonlinear in nature. In context, the linear data structure is a collection of datum items having similar types and the non-linear data structure is a type of data structure whereby datum elements are present at various levels [24].

In this regard, the use of artificial neural networks (ANNs) has increased considerably in recent years for the identification, classification and prediction of performance within soccer [25,26,27,28,29,30,31,32]. In particular, predictability has become possible according to players information through the analyses of data from past-recorded matches. For instance, the machine learning algorithms that have been employed within soccer research are K-Nearest neighbors [33,34], Neural networks [25,26,27,28,30,35,36], Decision Trees [37,38,39] and Bayesian networks [29,31,32,40,41].

There is limited research that has employed neural network approaches to data from the FIFA World Cup 2018. Specifically, Correa, et al. [42] used statistical simulations to analyze the progressive results of the tournament and predict the teams most likely to play in the finals and the team with the highest probability of becoming champions. In addition, Groll, et al. [43], investigated three different modelling approaches applied to data from the World Cups between 2002 and 2014 to predict performance in the 2014 World Cup. A hybrid random forest model, which combines random forests with Poisson ranking methods, was the best performing model. Lastly, Clemente [44], assessed the associations between different neural network measures such as group clustering or dyad reciprocity and performance outcomes (wins, goals scored and conceded and shots) to detect the variations within network measures between winners and losers. The results demonstrated that studied general networks were not sensitive enough to predict overall match outcome. Thus, as there is a dearth of research using player’s and team’s behaviors, the application of ANNs to predict World Cup performance is warranted.

The ease of availability of attributes/data from matches has meant that research using these performance indicators has become frequent. Artificial neural networks have become increasingly popular in the last few years to model and estimate characteristics that cause a team to win or lose a match, or predict the score of a particular match [22,45,46]. However, the modelling of match data to date has not been able to understand and interpret the vast amount of data available and simplify it in an objective manner. In addition, the difficulty of determining the actual contribution value of the complex attributes on match results, has meant that transferring such match attributes into the training process has been neglected so far in sport-related scientific literature. 

Therefore, the current study is the first to provide an application model of a learning neural network that can use official data analyzed by FIFA. The approach employed has not previously been used in the literature for match prediction. Further, there is novelty in the ability to determine the percentage effect of the matched attributes on the results objectively using the ANN model. 

The current study aimed to suggest a different approach to process the big data resulting from the use of devices, technological tools, and sensors to track and analyze football players ’performance. It was done by predicting the World cup 2018 match outcomes based on the match attributes that have been collected using FIFA technological tools during the match using a suggested neural network model. The second aim was to estimate the sensitivity of the match attributes affecting the chances of winning or losing using the neural network model.

## 2. Methods 

The concept of forecasting is to train the model to predict the match outcome based on the match attributes that have been collected after the match has finished. So, the study is carried out through the following steps:The match analyses.The ANN designing, training, testing.The match attributes’ sensitivities analysis.

### 2.1. Study Sample

Fifty-seven matches were officially analyzed by FIFA during the World Cup Finals 2018 using a recently validated TRACAB® system (position accuracy 0.09 m RMSE; total distance difference 0.42±0.60%) [47]. According to FIFA, the analysis provider used a real-time optical tracking system (25 frames per second including the player position coordinates *X*, *Y* in the playground as Pythagorean Theorem and the frame time) to collect the match attributes that constitute the input parameters of the ANN prediction model. Such a system has been successfully used in previous literature [1,13,15,48]. Seventy-five match attributes (from the official match score sheet) were selected to use in the ANN model. 

Those match attributes were from the physical indicators (distances covered, speed and sprints ... etc.) and technical indicators (number of passes, passes received, pass success, fouls, shots ... etc.) and are shown in the appendix. All matches that finished with 1 side wining were included (*n* = 55), matches that were drawn (*n* = 9) were not considered (*n* = 7) or were randomly selected for use in the validation phase (*n* = 2). The goals scored and conceded were not included due to their direct correlation with match outcome, which negatively affects the validity of the results. The match attributes nominal data were transformed into numerical values to be understandable for the ANN model, including technical and physical performance-related variables (see Appendix A), to quantify the match performance of the teams [46]. 

### 2.2. The ANN Model Construction

A commercial artificial neural network software NeuroDimension® [49], was used to predict and classify both teams in a match into two classes either win (class 1) or loss (class 2). The artificial neural network applied was a radial basis function (RBF neural network) [49]. Radial basis functions originated as a variant of the artificial neural network in 1988. Radial basis functions usually have 3 layers of the neural network; the input layer, a hidden layer that performs the functions of radial stimulation while the output layer applies the weighted hidden unit outputs. The input in the RBF network is nonlinear while the output is linear. So, RBF networks are able to model complex designations [50].

The construction of the RBF model relies on the design of the neural network elements, their arrangement, as well as the parameters that affect those [51]. The model contained three layers: an input layer, a hidden layer of processing elements (PEs) and an output layer as shown in Figure 1. 

The network standardizes the input data values to obtain a high accuracy using the root mean square error evaluation method (RMS) [52]. The Root mean square error (RMSE) is the square root of the mean square error (MSE). In the input layer, all match attributes are selected as input data while the desired data are coded to be 0 for a loss and 1 for the win. 

The ANN model cleans the input data automatically. The purpose of data cleaning is to remove noise, inconsistent data and errors in the training data [53]. The number of the input Processing Elements (PEs) was set to 75 (equal to the match attributes) and the output PEs was one (representing either win or lose). The weight regularization option was “weight decay,” with values often on a logarithmic scale between 0.01 and 0.001 as a default option.

The model later synthesizes the input data to create the RBF clusters using an unsupervised algorithm (Gaussian classifier) optimization process. Each cluster represents one RBF neuron and one type of playing variable. This neuron compared the similarity of its center with the input vector, with the amount of similarity representing the algorithm input value [54,55,56,57,58]. The competitive learning setting for each single PE is set as Standard Full. This allows the single-layer linear network to group and represent the match attributes in lie neighborhoods of the input space which are represented by a single output PE calculated using the weights according to Formula (1). The metric measures method was set as Euclidean [24].
(1)$$W_{2^{*}}\;\left(n+1\right)=W_{2^{*}}\;\left(n\right)+\eta \;\left(X\left(n\right)-W_{2^{*}}\;\left(n\right)\right)$$
where (*W*) is a weight from the synapse component that feeds the Gaussian Axon, (*η*) is the learning rate for each weight, (*n*) is the gradient and (*i**) is the PE that wins the competition.

The weights update mode is set to batch mode of the supervised learning. Data is randomly separated into three groups as training (65%), cross-validation (15%) and testing data sets (20%). The results were corroborated to each other by making the training; cross-validation and testing data sets include the same data ranges [19,59].

The increasing Mean Square Error (MSE) option was selected as termination during the cross-validation. This means the training iterations will automatically stop when the error value falls under the determined threshold value. Linear TahnAxon option was selected as an activation function interrelating the input and output parameters. The Levenberg–Marquardt (LM) algorithm was chosen as the learning rule to minimize the MSE of the ANN model. 

### 2.3. The ANN Training and Testing Procedures

The model was activated after its network building process was completed and first worked on the training data. After the training process phase, the test phase then started. The main aim of the test processing was determining the validity of the RBF model by activating it with the testing dataset. During the testing, the probe configuration was set as a prediction confusion matrix, providing lists of the real results against the prediction results for each class [60].

To show how the detection threshold in the RBF model affects detections versus false alarms the Receiver Operating Characteristic (ROC) matrix was chosen [49]. The RBF model estimates the error value of the model panel and minimizes the error as much as possible through the predefined amount of repetition and randomized weight values. The test report shows the percentage of success of the neural network model in identifying correctly the cases of a win or loss. Finally, sensitivity to the mean for all matches’ attributes were calculated to assess the effect of those attributes on the chance of winning or losing.

### 2.4. Statistical Analysis

To assess the training and testing process, the RBF training and testing report shows the RMSE which is the average of the errors between the target prediction and the real win or loss results. Additionally, Pearson correlation coefficients between predicted and target results were completed. 

## 3. Results

The differences between real and predicted loss and wins (RMSE) were 2.9% in the training phase and 0.37% in the testing phase. Differences in the RMSE in the loss and win RBF prediction model represent the different correct prediction percentage for each estimated probability versus the RBF model threshold. Pearson correlation coefficients between real and predicted loss and win were r = 0.72 for both. The correct percentages for the prediction model are 72.7% and 83.3% for loss and win, respectively. The estimated probability of positive prediction was used to evaluate the model creativity. The RBF predictor model makes a single prediction about each example win or loss. 

Receiver Operating Characteristic (ROC) matrices are used to show how changing the detection threshold affects the loss and win prediction detections versus false alarms. If the threshold is set too high then the RBF model will miss many detections. Conversely, if the threshold is set low then there will be too many false alarms (Figure 2). 

False Positive Rate is the percentage of false positives of those samples with an actual value of zero. True positive rate represents the percentage of true positives of those samples with an actual value of one. The false discovery rate is the percentage of false positives of those samples predicted as one. The best ROC threshold was 0.70 where there were no more false positive rates and the maximum true positive. 

Area under the ROC curve is the sum of the values that are displayed under the ROC chart and represents the overall performance of the RBF model. A measurement of 0.0 would be considered a poor model with no prediction power, while a measurement of 1.0 would be considered a best model [49,61]. As shown in the Figure 2A the area under curve is 0.91 for all positive rates where the acceptable true positive rate is 76.5% and false positive rate is 10%. Figure 2B shows the ROC area under curve for true/false positives rate. 

Based on the high correlation between the real and the predicted results, the radial basis function neural network (RBFNN) sensitivity analysis was produced by the software to assess the effect of match attributes on the loss or win prediction. These results indicated a valuable different contribution for each attribute on the wins and losses. (Figure 3).

## 4. Discussion

The key finding is that the prediction of match results based on ANN models provides an intelligent method to improve either the match events or the tactical training processes based on the relationship between game attributes and the outcomes in soccer. Previous literature has mainly focused on results prediction based on the location of the match (home or away), prediction dependent on a very low amount of match attributes or based on correlation coefficients between some attributes [45]. 

The present study aimed to present a method for the real application of such results in the training process and competition process using a wide range of match attributes associated with outcome. To the best of our knowledge, this is the first attempt to objectively quantify the match attributes sensitivity to improve match results or training procedures. ANN approaches for match results prediction have been previously used in research with either limited game attributes or using attributes that are not outcomes from the analysis, for example players’ injuries, weather conditions etc. employing ANN and logistic regression (LR) techniques [45]. In contrast with this research, the current study employed an RBF neural network model and successfully predicted the team’s results as either loss or win 72.7% and 83.3% of the time, respectively.

The prediction percentage is comparable with the results of other studies. Igiri and Nwachukwu [62], used a data mining tool (Rapid Miner) to predict the results of soccer matches with 75.04% prediction accuracy when the ANN technique and logistic regression were used to build the model. Others have used the Markov Process Model and their results yielded an accuracy of 56% [63]. While the previous studies discuss several merged methods to predict and classify team results, they did not discuss the Estimated Positive Probability, which explains the probability that the network output equaling the ROC Detection Threshold will have a desired output of one. The ROC curve is used to examine the tradeoff between the detection of true positives while avoiding false positives. In Figure 2 the RBF model has the best threshold of 0.70 where the best detection of true positive rate was 76.5% versus false positives of 10% and the best maximum difference rate was 66.5%.

The relationship between sports results and various data elements are directly affected by different tactics and techniques factors. The sensitivity analysis based on ANN have been suggested to predict the results based on available data. Therefore, the employment of the ANN will provide a more objective and credible method to predict performance compared with relying solely on the experience and instincts of experts, which include a high error ratio, or relying on basic statistical data. The ANN sensitivity analysis for players performance and matches attributes will result in more reliable predictions that support the coach and trainer decision-making [64]. 

The success of the ANN model in match results prediction and in establishing a relationship between the attributes and matches results correspond to the findings of some previous studies except with the value of the attributes’ sensitivity [65,66,67]. The second contribution of using the ANN model is the possibility to draw the effect of each attribute on the results of the matches as shown in Figure 3. The attributes were divided into four groups based upon how powerful they were in predicting success (≤25%, ≤50%, ≤75% and ≤100%). The most powerful attributes for predicting success were the Total Team Medium Pass Attempted (MBA) at 100%, the Average of Distance Covered by the Team where the player’s speed ranged between 15 and 20 km/h (Zone3_TA) at 99%, the Team Average ball delivery into the attacking third (TA_DAT) at 80.9%, the Total Team Covered Distance without the Ball (Not_in_Poss_TT) at 76.8% and the Average Distance Covered by the Team (Game_TA) at 75.1%.

These results indicate that teams who employ medium distance passes to maintain possession are more successful. This finding is consistent with previous research that confirmed that the teams who control the game by passing, can patiently move the ball to search for defensive weaknesses [68]. In addition, our results are consistent with previous studies conducted on the same competition, where it was stated that the teams that qualify for the knockout stage are distinguished by their possession of the ball with their direct play and handing the ball in the offensive third of the field [46]. Furthermore, our analysis using artificial neural networks indicated that the distance covered by the team without possession of the ball and the average distance covered by the team as a whole were key attributes for success reaching 76.8% and 75.1%, respectively. This is also consistent with previous research that confirmed that successful teams covered a greater distance at a low speed and maintained possession of the ball with moderate passes [46]. Much of the previous research has focused primarily on high-intensity running with limited attention to low-intensity activity [69,70], perhaps they consider it as a measure of players’ physical abilities [71]. Such ANN analysis allows the rationalization of training and provides a deeper understanding of the management of the game, technically and strategically.

The study has limitations as the isolated data from the physical parameters do not allow the control for situational variables, and a drawback of the method presented in this study is the time needed to obtain the match data. The FIFA determination of the match attributes might only be available sometime after the completion of the match or competition. However, the latest TRACAB system does provide data in real-time so if this can be accessed, by the sports scientist, the ANN could be employed the same day allowing feedback between matches. Therefore, the RBF model methods presented for prediction and sensitivity analysis could simplify and improve the training process significantly.

As soon as the match data is available, the calculation of the RBFNN is done in a very short time. Moreover, calculated attributes contributions will be estimated accurately. However, the error due to the sensitivity model will be acceptable when we consider the effective attributes in terms of the match results. Another challenge might be the difficulty for trainers to use ANNs. Performance analysts and statisticians must be qualified to use such software created by sport scientists in cooperation with computer specialists to optimize the benefits available to enhance soccer performance. Comparison with the few papers that have focused on such an approach is not possible as the rigorous statistical analysis of the important variables was not similar. Future research needs to assess and determine the efficacy of the ANN presented in enhancing performance and to employ a multivariate and integrative approach with physical, positional, technical and tactical variables.

## 5. Conclusions

In the current study, we provide an approach to predict the win, loss and the sensitivity of match attributes from the soccer World Cup 2018. Using the commercial software, NeuroDimension® (NeuroDimension, Gainesville, F.L., USA) [49], a supervised neural network based on Radial basis function using 75 attributes presented in the official match report by FIFA for the World Cup was produced. The match attributes were selected and trained within the network training model. The match results were selected as a target output. After the neural network training phase finished the test phase was performed. About 72.7% of the loss results could be predicted as desired results, while 83.3% could be predicted as desired win results. In order to optimize the benefit of analysis of the attribute, the match attributes’ sensitivities were analyzed. The original neural network model successfully presented the importance of each attribute for the match results. Summarizing, the study clearly demonstrates that the novel neural network can be used to predict results of matches, effective attributes and consequently useful to support trainers’ and coach’s decisions during the match and improve tactical training in team soccer and similar types of sports. It offers an objective approach to manage training and matches pertaining to match attributes’ sensitivities and thus avoids coach’s self-impressions.

## 6. Practical Applications

This study provides a deep insight into the effects of match attributes on the outcome of matches. It is also possible to understand the interrelationship between different match variables that are difficult to find statistically. The medium passes and delivery into the attacking third in addition to the movement without the ball and the running at medium speeds have proven important attributes that determine the results of matches, in agreement with similar studies. Depending on the important matching features as strong or weak forecasting factors, new tactical play methods can be identified and created, or against the perceptions presented by the competitor, to ensure that the best physical and technical capabilities of the players are employed. Considering the limitations of the study, it is possible to monitor the important matching features in the training sessions or matches and re-evaluate the nature of the performance and tasks assigned to the players in order to develop the style of play. Results must be taken with caution because of the small sample size from the FIFA World Cup. This restriction also imposes further research to consider proposing other tools and means that contribute to exploring and examining play situations and their relationship to physical performance and match characteristics with strong or weak influence on the outcome of matches alike.

## Figures and Tables

**Figure 1 sensors-20-03213-f001:**
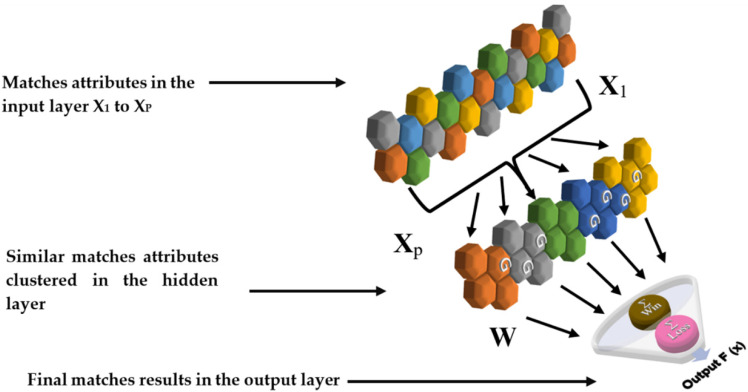
Radial Basis Function (RBF) network. G is Gaussian function, W is weights.

**Figure 2 sensors-20-03213-f002:**
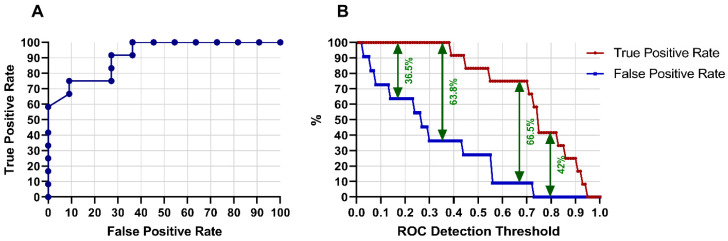
(**A**): Illustrates receiver operating characteristic (ROC curve) and accuracies through thresholds after 100 training iterations improvements. For the testing sets, with pedestrial/non-pedestrial threshold = 1, RBF neural network predicts correctly 91%. (**B**): Comparison between the true and false positive prediction rate.

**Figure 3 sensors-20-03213-f003:**
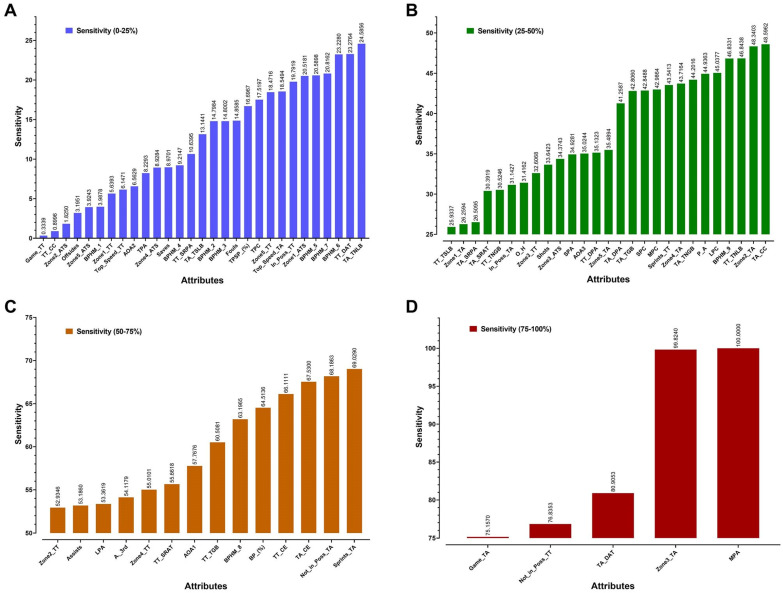
Contribution of match attributes to results (ANN sensitivity analysis); (**A**): sensitivity from 0 to 25%, (**B**): sensitivity from 25 to 50%, (**C**): sensitivity from 50 to 75%, (**D**): sensitivity from 75 to 100%.

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
