# Peer review of "Predicting Wins, Losses and Attributes’ Sensitivities in the Soccer World Cup 2018 Using Neural Network Analysis"

_sensors, 2020, doi:10.3390/s20113213_

Round 1

Reviewer 1 Report

The authors provide an approach to predict the win, loss and the sensitivity of match attributes from the soccer World Cup 2018. Using the commercial software NeuroDimension, a supervised neural network based on RBFs using 75 attributes presented in the official match report by FIFA for the World Cup was produced.

The paper is well-written in general, but it is not always self-contained: Some notions are not explained in the paper, nevertheless references are given.

The root-mean-square error of 2.9% looks good, but it seems that this is only the training error. If this is the case, the authors should say what is the test error. If the test error is much higher, then the paper should not be published to my opinion. If the test error is 2.9%, then the paper can be accepted after some revision.

Specific remarks:

p.1,l.20: What is meant by "linear and non-linear data"? Please explain at least by a small paraphrase!

p.1,l.28: Is 2.9% the training or the test error? This should be clarified already in the abstract. See also my comment below.

p.2,l.59: For reference [33], the author names are listed. To my opinion they should be omitted like at all other references.

p.3,l.98: What does "player position coordinates X, Y in the playground as Pythagorean Theorem" mean? How is the Pythagorean theorem applied in this context?

p.3,l.114: What is an RBF? How are they used in the neural network? I would expect a brief definition here.

p.3,l.120: How is the RMS defined? To make the paper more self-contained, I would expect a brief definition also here.

p.4,l.137: Formula 1 needs more explanation: What is W_2*? What is the meaning of the \cap symbol here, or is this a typing error?

p.4,l.168: "In the training phase, the differences between real and predicted loss and wins (MSE) were 2.9%." The root-mean-square error of 2.9% looks good, but it seems that this is only the training error. If this is the case, the authors should say what is the test error. The experimental setting obviously contained test data (cf. p.4,l.140). A small training error is not surprising in neural network learning.
Furthermore, is 2.9% the root-mean-square error (as written in the abstract) or is it the MSE (i.e., without taking the square root)? This should be clarified. In addition, there should not be a line break between "2.9" and "%".

p.7,l.242: How much time is needed to obtain the match data? Maybe it can be discussed here whether the system could be turned into a real-time system (in future work).

Reviewer 2 Report

  • Abstract, please add more information about the key variables and situational variables that were identified as powerful predictors of success.
  • Keywords, please use different words than title for example: football instead of soccer, match analysis instead of soccer analysis, sport analytics instead of neural network.
  • Introduction: this section could include more references that were done focused on predicting match outcome in soccer:

Casamichana, D., Castellano, J., Galleja-González, J., & San Román, J. (2013). Differences between winning, drawing and losing teams in the 2010 World Cup. Science and Football VII, London: Routledge, 211-216

Castellano, J., Casamichana, D., & Lago, C. (2012). The use of match statistics that discriminate between successful and unsuccessful soccer teams. Journal of human kinetics31, 137-147.

Harrop, K., & Nevill, A. (2014). Performance indicators that predict success in an English professional League One soccer team. International Journal of Performance Analysis in Sport14(3), 907-920

Lago-Ballesteros, J., & Lago-Peñas, C. (2010). Performance in team sports: Identifying the keys to success in soccer. Journal of human kinetics25(1), 85-91

Lago-Peñas, C., Lago-Ballesteros, J., & Rey, E. (2011). Differences in performance indicators between winning and losing teams in the UEFA Champions League. Journal of human kinetics27(1), 135-146

Lago-Peñas, C., Lago-Ballesteros, J., Dellal, A., & Gómez, M. (2010). Game-related statistics that discriminated winning, drawing and losing teams from the Spanish soccer league. Journal of sports science & medicine9(2), 288

Zhou, C., Zhang, S., Lorenzo Calvo, A., & Cui, Y. (2018). Chinese soccer association super league, 2012–2017: key performance indicators in balance games. International Journal of Performance Analysis in Sport18(4), 645-656

  • Introduction section should introduce the importance of contextual-related factors in soccer. These aspects highly modify the soccer teams’ performance during matches and competitions.

Bradley, P. S., Lago-Peñas, C., Rey, E., & Sampaio, J. (2014). The influence of situational variables on ball possession in the English Premier League. Journal of Sports Sciences32(20), 1867-1873.

Lago-Ballesteros, J., Lago-Peñas, C., & Rey, E. (2012). The effect of playing tactics and situational variables on achieving score-box possessions in a professional soccer team. Journal of Sports Sciences30(14), 1455-1461

  • Introduction, L49, please clarify the importance of multivariate analyses and machine learning when analyzing soccer match performance. I suggest some readings:

Moura, F. A., Martins, L. E. B., & Cunha, S. A. (2014). Analysis of football game-related statistics using multivariate techniques. Journal of sports sciences32(20), 1881-1887

Robertson, S. (2015). Games by numbers: machine learning is changing sport. Retrieved from: http://theconversation.com/games-by-numbers-machine-learning-is-changing-sport-38973 (28-09-2016).

  • Introduction, please differentiate the analyses using networks in soccer: (i) social network; (ii) neural network and (iii) network science. Please use the following references to justify the rationale:

Buldú, J. M., Busquets, J., Martínez, J. H., Herrera-Diestra, J. L., Echegoyen, I., Galeano, J., & Luque, J. (2018). Using network science to analyse football passing networks: Dynamics, space, time, and the multilayer nature of the game. Frontiers in psychology9, 1900

Buldu, J. M., Busquets, J., & Echegoyen, I. (2019). Defining a historic football team: Using Network Science to analyze Guardiola’s FC Barcelona. Scientific reports9(1), 1-14.

Martínez, J. H., Garrido, D., Herrera-Diestra, J. L., Busquets, J., Sevilla-Escoboza, R., & Buldú, J. M. (2020). Spatial and Temporal Entropies in the Spanish Football League: A Network Science Perspective. Entropy22(2), 172

  • Methods: I am concerned about the use of physical data without explaining the data validity and reliability. Please add this required information to this section.
  • Physical data associated to winning/ losing: I consider that the analyses should be related to the contextual-related factors. Why the authors did not use these variables.
  • I have surfed the web to seek some FIFA articles related to win lose and physical/technical demands. Then, I identified some references that need to be considered for your rationale and method, for example controlling for the style of play of each team or the match type.

Yi, Q., Gómez, M. A., Wang, L., Huang, G., Zhang, H., & Liu, H. (2019). Technical and physical match performance of teams in the 2018 FIFA World Cup: Effects of two different playing styles. Journal of sports sciences37(22), 2569-2577

  • Figure 3 is unclear at best. I am not able to identify each attribute. I strongly encourage the authors to clearly explain the key attributes (instead of codes) identified as relevant or powerful.
  • As a sport scientist I am interested in the main results that can be applied into practice. Then, the authors should rewrite the section clarifying that issue.
  • Discussion: please use the references suggested above to improve the meaning and arguments used in discussion section. In addition, the authors should include the limitations of this study (eg. Isolated data from physical parameters, the need to control for situational variables, etc.) and further research (need of multivariate and integrative approach with physical, positional, technical and tactical variables).
  • I need to read at the end of the article the real practical applications obtained from this study. If no real practical tasks or information can be identified, the article does not mean anything. Please keep this issue in mind to rewrite the manuscript accordingly.
  • Appendix: as I suggested above, the authors need to define each variable/ indicator in order to clarify the variables analysed. Without that description, the results are hard to read.

Round 2

Reviewer 1 Report

As far as I can see, the authors incorporated the reviewer comments thoroughly. The paper is now significantly more self-contained. In addition, they made clear that the testing error is rather low. The paper thus can be published to my opinion. I have only two comments:

p.4,l.141-142: "The Root mean square error (RMSE) is the square root of the mean square error (MSE) and is the sum of the variance and the square bias." I find the second part of the sentence a bit misleading. It may be omitted. The RMSE can be seen as the square root of the variance, not their sum.

p.5,l.193-194: "In the training phase, the differences between real and predicted loss and wins (RMSE) were 2.9% and in the testing phase was 0.37%." I would re-arrange the words as follows: "The differences between real and predicted loss and wins (RMSE) were 2.9% in the training phase and 0.37% in the testing phase."

Author Response

Detailed Response to the Editor and Reviewers

Dear Prof. Alison Zhang, dear reviewers,

We thank the editor and the reviewers for the careful and critical reading of our manuscript.

We completely agree with their comments and suggestions that improved the quality of the manuscript, following their remarks, a summary of the corrections/modifications enclosed in the revised version.

We have modified the manuscript accordingly, and detailed corrections are listed below point by point.

First reviewer comments:

1

Comment

p.4,l.141-142: "The Root mean square error (RMSE) is the square root of the mean square error (MSE) and is the sum of the variance and the square bias." I find the second part of the sentence a bit misleading. It may be omitted. The RMSE can be seen as the square root of the variance, not their sum.

Response

The second part of the sentence was deleted.

Now it is only:

The Root mean square error (RMSE) is the square root of the mean square error (MSE).

2

Comment

p.5,l.193-194: "In the training phase, the differences between real and predicted loss and wins (RMSE) were 2.9% and in the testing phase was 0.37%." I would re-arrange the words as follows: "The differences between real and predicted loss and wins (RMSE) were 2.9% in the training phase and 0.37% in the testing phase."

Response

The words re-arranged, now it is only:

The differences between real and predicted loss and wins (RMSE) were 2.9% in the training phase and 0.37% in the testing phase.

Regards

Authors

Reviewer 2 Report

The authors addressed satisfactorily the reviewer's comments

Author Response

Dear reviewer, 

Since there are no more modifications required, we thank you so much for the careful revision of our manuscript.

Authors